# Rapid adaptive evolution of avian leukosis virus subgroup J in response to biotechnologically induced host resistance

**Magda Matoušková**[ORCID]**, Jiří Plachý, Dana Kučerová, Ľubomíra Pecnová, Markéta Reinišová, Josef Geryk, Vít Karafiát, Tomáš Hron, Jiří Hejnar**[ORCID]*

Department of Viral and Cellular Genetics, Institute of Molecular Genetics, Czech Academy of Sciences, Prague, Czech Republic

* jiri.hejnar@img.cas.cz

**Data Availability Statement:** All data are in the manuscript and its supporting information files.

## Abstract

Genetic editing of the germline using CRISPR/Cas9 technology has made it possible to alter livestock traits, including the creation of resistance to viral diseases. However, virus adaptability could present a major obstacle in this effort. Recently, chickens resistant to avian leukosis virus subgroup J (ALV-J) were developed by deleting a single amino acid, W38, within the ALV-J receptor NHE1 using CRISPR/Cas9 genome editing. This resistance was confirmed both *in vitro* and *in vivo*. *In vitro* resistance of W38$^{-/-}$ chicken embryonic fibroblasts to all tested ALV-J strains was shown. To investigate the capacity of ALV-J for further adaptation, we used a retrovirus reporter-based assay to select adapted ALV-J variants. We assumed that adaptive mutations overcoming the cellular resistance would occur within the envelope protein. In accordance with this assumption, we isolated and sequenced numerous adapted virus variants and found within their envelope genes eight independent single nucleotide substitutions. To confirm the adaptive capacity of these substitutions, we introduced them into the original retrovirus reporter. All eight variants replicated effectively in W38$^{-/-}$ chicken embryonic fibroblasts *in vitro* while *in vivo*, W38$^{-/-}$ chickens were sensitive to tumor induction by two of the variants. Importantly, receptor alleles with more extensive modifications have remained resistant to the virus. These results demonstrate an important strategy in livestock genome engineering towards antivirus resistance and illustrate that cellular resistance induced by minor receptor modifications can be overcome by adapted virus variants. We conclude that more complex editing will be necessary to attain robust resistance.

## Author summary

The CRISPR/CAS9 technology accelerates genetic modification studies. Induction of the farm animals' virus resistance is particularly important. However, the high adaptability of viruses could present a serious drawback to this effort. Their adaptive evolution in hosts with biotechnologically induced resistance has received minimal attention so far. Using a single amino acid deletion in the specific receptor we previously created a chicken line

**Funding:** J.H. received funds from the Czech Academy of Sciences (Praemium Academiae, and the program Strategy AV21, project Virology and Antiviral Therapy) and the National Institute of Virology and Bacteriology (Program EXCELLES, ID LX22NPO5103 - funded by the European Union - Next Generation EU). The funders had no role in the study design, data collection and analysis, decision to publish, or preparation of the manuscript.

**Competing interests:** The authors have declared that no competing interests exist.

resistant to avian leukosis virus subgroup J. As little change as possible is desirable to preserve gene function and minimize any side effects on the fitness of the chickens. Here we however show that the virus rapidly adapts to the single amino acid deletion in the receptor while the cells with extensive receptor changes remain resistant. Our study highlights the need for complex gene editing and validating the resistance to minimize the risk of virus evolution and adaptation to the genetically modified host.

## Background

Gene editing technologies, specifically clustered regularly interspaced short palindromic repeats commonly known as CRISPR, have the potential to revolutionize agriculture by implementing targeted modifications into the genomes of crops and domestic animals. Of particular importance are the new strategies utilized to increase resistance to viruses in livestock by targeting specific genes that are involved in virus entry or replication [1]. Until now, this approach has been successfully employed in the preparation of porcine reproductive and respiratory syndrome virus-resistant pigs [2], avian leukosis virus (ALV)-resistant chickens [3,4], and avian influenza-resistant chickens [5,6]. The resistance to avian leukosis virus subgroup J (ALV-J) has been achieved by a single amino acid deletion in a specific cell receptor and represents a proof of concept for similar strategies to combat other poultry diseases.

ALV is a complex of closely related virus subgroups that have evolved through mutations and recombinations in the sequence of the envelope (Env) protein, which is crucial for specific receptor binding and cell entry. Mutations occur primarily within the surface unit (SU) in the variable regions vr1, vr2, and vr3 and host range determining regions hr1 and hr2 of the receptor binding site [7–10]. ALV has diversified into subgroups with different receptor usage and distinct range of susceptible hosts. Currently, ten subgroups of ALV are defined.

Among the known ALV receptors, there are cell surface proteins such as the cobalamin transporter Tva [11–14], a member of the tumor necrosis factor receptor family Tvb [15,16], the butyrophilin-like protein Tvc [17], and Na+/H+ exchanger type 1 (NHE1) [18], which all exhibit various functions. Different ALV subgroups bind different receptor molecules or different binding sites on the same molecule. The plausible evolutionary scenario of such receptor diversification, the virus-host arms race, assumes not only virus adaptation but also anti-virus immunity, development of virus restriction by the host, and resistance at the receptor level. In regards to receptor resistance, the presence of multiple alleles encoding virus-resistant receptor variants in domestic poultry and in wild galliform birds has been described [17,19–24].

ALV-J has emerged through the recombination of an unknown exogenous ALV and an endogenous EAV/HP retrovirus element [25–29]. The evolution of the virus accelerated after the transmission from broilers to laying chicken breeds [30] and ALV-J is now highly diversified [31]. The env gene variations were shown to be particularly important for virus adaptation [30,32,33]. ALV-J remains a serious concern for the poultry industry in south-east Asian countries [25]. It uses the housekeeping chicken gene NHE1 as a receptor for entering target cells [18], and domestic chickens, jungle fowls, turkeys, and New World quails are the only susceptible species [24,34]. A comparison of the NHE1 amino acid sequence in ALV-J susceptible and resistant species pointed to W38 as a residue critical for receptor function [24]. Since polymorphisms in W38 are not present in chicken breeds [35], the resistance to ALV-J was prepared biotechnologically *in vitro* by introducing either frame-shifting indels or in-frame deletions including W38 [23,36]. To prepare *in vivo* resistance to ALV-J, the deletion of single amino acid W38 was performed which posed minimum risk of NHE1 malfunction [3].

On the other hand, minimizing the change of the target protein poses a risk of virus adaptation. Therefore, we have used our simplified ΔW38 NHE1 system to explore the mechanisms of virus adaptation to an altered host cell receptor in this study. This approach not only improves our understanding of virus-host coevolution but is also instructive for future biotechnological derivation of pathogen-resistant livestock.

## Results

### Selection of ALV-J variants adapted to ΔW38 NHE1

The experimental scheme and overall workflow is depicted in Fig 1A. The RCASBP(J)GFP vector was used to model ALV-J infection in chicken embryonic fibroblasts (CEF). RCASBP(J) GFP is an ALV-based replication competent vector of subgroup specificity J that transduces

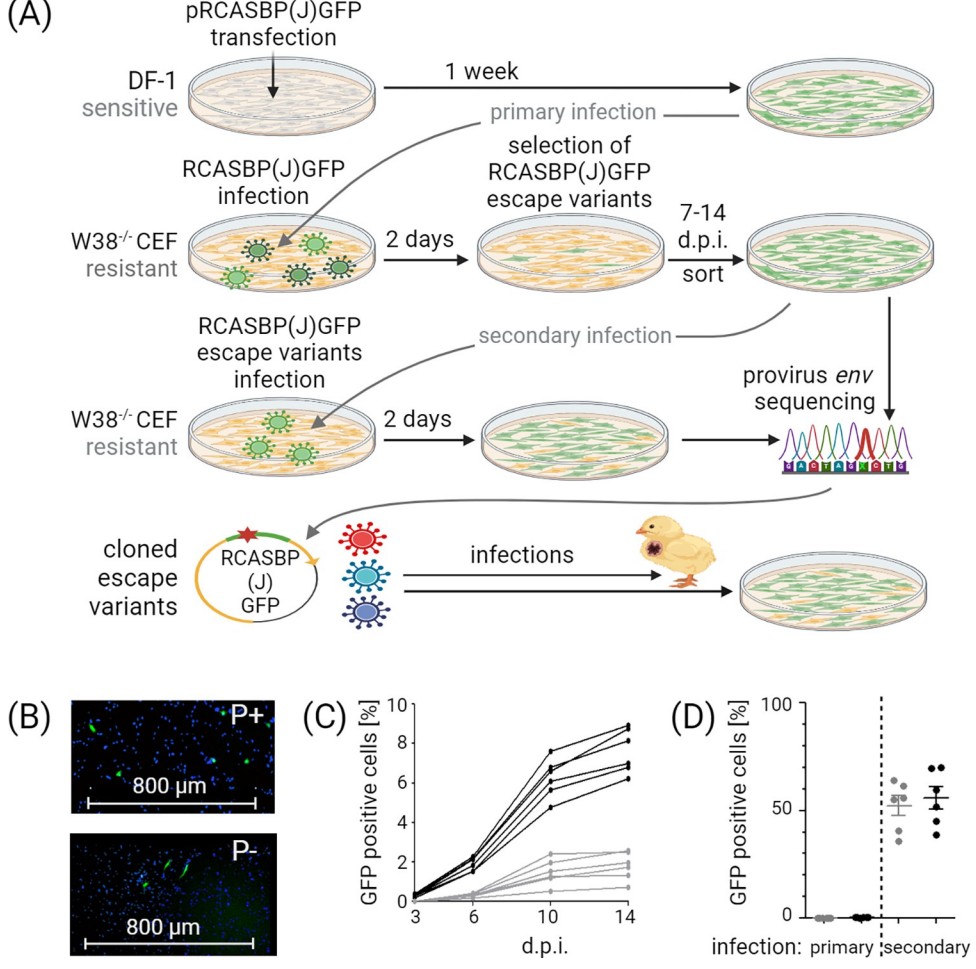

**Fig 1. Detection of adapted RCASBP(J)GFP variants that overcome W38$^{-/-}$ CEF resistance.** (A) Preparation of adapted variants: schematic representation of the workflow and timeline. (B) Representative images of W38$^{-/-}$ fibroblasts observed by fluorescence microscope 2 d.p.i. with (P+) or without (P-) polybrene support are shown. Cell nuclei were stained with DAPI. (C) Outcome of the primary infection: six RCASBP(J)GFP virus stocks were prepared by transfection of DF-1 cells with pRCASBP(J)GFP plasmid in six parallels. Virus was collected 1 week after transfection. W38$^{-/-}$ CEF infections performed either with support of polybrene (black lines) or without (gray lines) are shown. The virus spread in the cell culture was assayed as the percentage of GFP-positive cells by FACS for 14 days. (D) Comparison of W38$^{-/-}$ CEF primary and secondary infection efficacy with (black dots) or without (gray dots) polybrene support 2 d.p.i. Figure was created with Biorender.com.

GFP (Fig 2A). The RCASBP(J)GFP virus was propagated in DF-1 cells in replicates of six and was then used to infect CEF derived from W38[-/-] chickens [3] with a multiplicity of infection (MOI) 5.0. In order to increase the efficiency of infection, we also performed parallel infection with addition of polybrene. Two days post infection (d.p.i.), individual GFP-positive cells were observed in the cultures via fluorescence microscopy (Fig 1B). Importantly, the percentage of GFP-positive cells 3 d.p.i. was at the level of background. Unlike in experiments performed in a previous study [3], we kept the cell culture for an additional two weeks and monitored GFP positivity by flow cytometry (Fig 1C). Within two weeks after the primary infection, the percentage of GFP-positive cells had reached 9% in the presence of polybrene in comparison to 2% to cultures without polybrene (Fig 1C). GFP-positive cells were sorted at 7–14 d.p.i. and cultured for an additional 7–14 days. The titer of the GFP-transducing virus in media collected from sorted cells was significantly increased and the percentage of GFP-positive W38[-/-] CEF averaged was more than 50% three days after secondary infection regardless of the addition of polybrene (Fig 1D). This experiment suggests rapid adaptation of the virus to host resistance induced by a single amino acid deletion.

## Detection of amino acid substitutions in Env of adapted ALV-J variants

Because the retroviral Env plays a decisive role in virus entry, we expected adaptive changes to occur within the *env* gene. We amplified and sequenced the *env* gene of RCASBP(J)GFP present in GFP-positive W38[-/-] CEFs after primary and secondary infection (Fig 1A). Apart from vector stock preparation and infection presented in previous experiments, more samples for sequencing were prepared independently in a similar manner (see S1 Table for details). From CEFs infected with and without polybrene eight different substitutions were detected in RCASBP(J)GFP *env*: S236L, N311S, N311D, N311K, T313A, T313I, T313N, and A432T (Fig 2A). At least one mutation was detected in each sample of the infected W38[-/-] CEFs. Most of the observed mutations appeared recurrently in W38[-/-] CEFs infected with independent vector stocks. Similar mutations were observed in infected cells regardless of the addition of polybrene (S1 Table). We did not observe any prevalent mutations in the infected DF-1 cells. Remarkably, all mutations occurred outside the variable and host-range regions. Most of the observed mutations disrupted one particular N-glycosylation site, NxT [37], within the SU at either an asparagine or threonine. S236L is situated within SU upstream from hr2, A432T is located in the heptad repeat 1 (HR1) and was the only mutation in the transmembrane unit (TM). In available Env of ALV-J (Env-J) sequences (Dataset S1, sheet A) [31], the substituted amino acids are rather invariable. Based on the ratio of non-synonymous to synonymous nucleotide substitutions, the diversifying selection of the substituted amino acid sites was not detected (Dataset S1, sheet B). As suggested by AlphaFold modeling, all mutations are located on the surface of the Env glycoprotein molecule and, in particular, the glycosylation site N311 with the protruding glycan and T313 are on the top of Env where they are exposed for receptor binding (S1 Fig).

S236L, T313A, T313I, and A432T are substitutions that change the hydrophobicity of the side chain, while N311D and N311K are substitutions that change the charge of the side chain.

To assess the role of detected mutations, each of them was separately introduced into the RCASBP(J)GFP vector. Surprisingly, all eight new virus variants were able to overcome W38[-/-] CEF resistance and retained their infectivity in DF-1 cells (Fig 2B and 2D). The adapted virus variant harboring an A432T substitution within HR1 infected W38[-/-] fibroblasts less effectively than all other adapted mutants (Fig 2B). Pairwise statistical comparisons of all variants are shown in S2 Table. The absolute percentage of GFP-positive cells 2 d.p.i. varied from 40 to 80% in W38[-/-] CEF and from 49 to 86% in DF-1 cells.

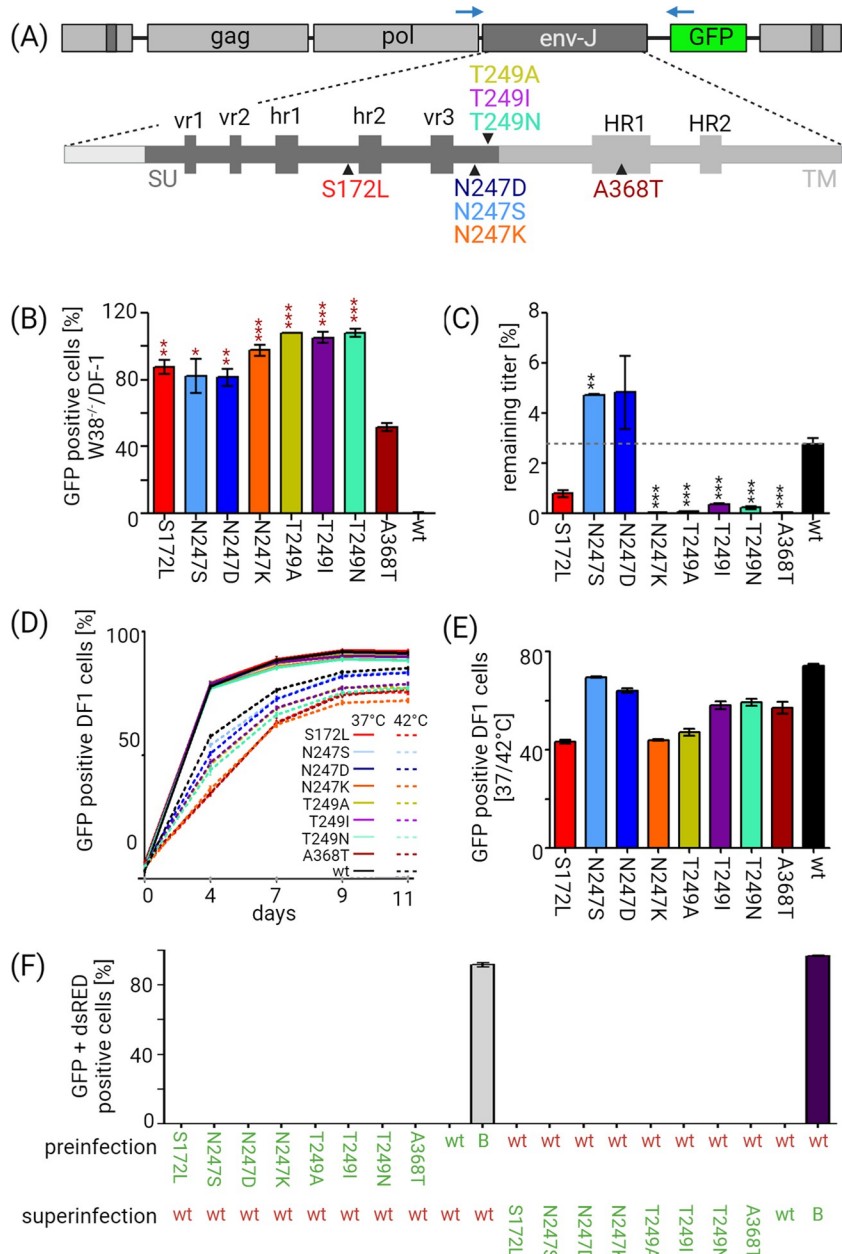

**Fig 2. Characterization of adapted ALV-J variants.** (A) Scheme of RCASBP(J)GFP and Env glycoprotein domain structure with positions of all amino acid substitutions detected in adapted variants (black triangles). Host-range regions (hr1/2) and variable regions (vr1/2/3) in the SU are depicted as dark gray boxes. Heptad repeats (HR1/2) in the TM are shown as light gray boxes. *env* was amplified by depicted primers (blue arrows) from RCASBP(J)GFP infected W38$^{-/-}$ fibroblasts. (B) W38$^{-/-}$ CEFs and DF-1 cells were infected with wt RCASBP(J)GFP and eight cloned variants each carrying a single adaptive mutation. Cells were analyzed 2 d.p.i. by FACS. The percentage of GFP-positive W38$^{-/-}$ fibroblasts was normalized to the percentage of GFP-positive DF-1 cells infected in parallel. Each infection was performed in triplicate and means ± SD are shown. Student's T-test comparison is shown for comparison with the A432T variant. T-test comparisons for all variants are shown in Supplementary Information (S2 Table). (C) Thermosensitivity of the RCASBP(J)GFP variants. Triplicates of RCASBP(J)GFP variants were incubated at 42°C or kept on ice for 3 hours. DF-1 cells were infected with these viruses and the GFP positivity was determined 3 d.p.i. by FACS. The results are presented as the ratio of GFP-positive DF-1 cells infected by the virus kept at 42°C and GFP-positive DF-1 cells infected by the virus kept on ice. Each value represents the mean percentage ± SD. (D) Virus spread at 37 and 42°C. DF-1 cells were infected with all virus variants and GFP positive cells were sorted. The infected cells were mixed with fresh DF-1 cells in the ratio 1:19. The virus spread was calculated as the percentage of GFP-positive cells by FACS at 4, 7, 9, and 11 days of co-cultivation. (E) Difference of virus spread in DF-1 cells at 42°C normalized

to the percentage of GFP positive cells cultured at 37°C. The GFP positivity was measured 4 days after mixing with fresh cells. T-test comparisons for all variants are shown in supplementary information (S3 Table). (F) Superinfection interference of RCASBP(J)GFP variants (green labels of substitutions) with the wt RCASBP(J)dsRED (red wt). RCASBP(B)GFP (green B) was used as a negative control of infection interference. The results are shown as mean percentages of GFP-positive/dsRed-positive cells measured in triplicate. Figure was created with Biorender.com.

In order to determine whether these mutations enable the virus to adapt to the modified NHE1 receptor or enable it to use another molecule to enter the cell we have performed the interference assay based on competition for receptor and entry interference between viruses of the same subgroup. We preinfected DF-1 cells either with all adapted variants of RCASBP(J) GFP and, two weeks later, we challenged the preinfected cells with wt vector carrying a dsRed fluorescent marker (RCASBP(J)dsRED). We also performed the reciprocal assay with wt RCASBP(J)dsRed as preinfection virus and adapted variants of RCASBP(J)GFP as a challenge. FACS analysis showed no evidence of GFP/dsRED double positive cells in the superinfected DF-1 culture. This strong interference suggests that the adapted variants use the same receptor NHE1 (Fig 2F). As a control, we used the B subgroup RCASBP(B)GFP vector, which did not interfere with any of the J subgroup viruses. Although we cannot completely rule out weak affinity for novel receptors or receptor-independent entry into cells, strong interference indicates that this possibility is unlikely.

Adaptive mutations in the env gene are often accompanied by a reduction in viral fitness, e.g. slower replication, sensitivity to pH fluctuations, and reduced thermal stability. The temperature instability may be biologically relevant because the body temperature of a chicken is 3–5°C higher than the temperature used for the propagation of the virus *in vitro*. The infectivity of all RCASBP(J)GFP variants, including the wt, substantially decreased after 3 hours at 42°C. In comparison to wt, the variant with the N311S substitution is even more thermostable, variants with N311D and S236L substitutions are slightly but statistically insignificantly different, while all other variants were found to be distinctly thermosensitive (Fig 2C). The spread of the virus was slower at 42°C than at 37°C (Fig 2D). Furthermore, while all virus variants replicated with the same efficiency at 37°C, the replication efficiency at 42°C varied (Fig 2D and 2E) and the thermostable variants N311S,D replicated better than variants proven as temperature-sensitive in the previous experiment (Figs 2D and S3).

Altogether, these experiments showed that single amino acid substitutions within Env-J are sufficient enough to circumvent ΔW38-induced resistance. Most of the observed substitutions emerged recurrently. Additionally, all viruses replicated efficiently in DF-1 and W38$^{-/-}$ CEFs and varied mostly in regards to the thermosensitivity of a variant.

## Transforming virus pseudotyped with adapted Env-J variants induces tumors in W38$^{-/-}$ chickens

Next, we tested the sensitivity of W38$^{-/-}$ chickens to each variant by inducing tumors with a replication-defective v-*src*–transducing virus without *env* gene, which had been pseudotyped with adapted variants of Env-J. Since the natural body temperature of a chicken varies between 41 and 42°C, we selected the variants that showed the highest stability at 42°C *in vitro* (N311S, N311D, and S236L substitutions) for *in vivo* experiments. We compared these stable variants to the least thermostable variant that contains an A432T substitution and employed the wt Env-J variant as a control. The titer of the transforming pseudotyped virus was determined by an *in vitro* focus assay and measured as focus-forming units (FFUs) on CEF cells. Each chicken in this experiment was inoculated with 250 FFU into the left side and 500 FFU into the right side of the breast muscle. A minimum of four W38$^{-/-}$ and three W38$^{+/+}$ chickens aged 19 to 34 days for each ALV-J variant were used for the experiment.

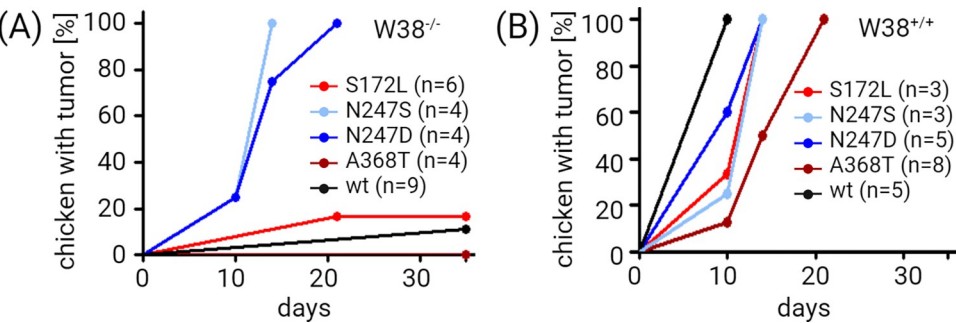

**Fig 3. Sensitivity of W38<sup>-/-</sup> chickens to selected adapted RCASBP(J)GFP variants.** V-*src*-transducing virus pseudotyped with selected variants of J envelope was used to induce tumor growth in W38<sup>-/-</sup> (A) and in control W38<sup>+/+</sup> chickens (B). Tumor growth was inspected 10, 14, 21 and 35 d.p.i. Each dot represents the percentage of chickens infected with a particular virus variant carrying a tumor at the time of inspection. W38<sup>-/-</sup> chickens and control W38<sup>+/+</sup> chickens were infected by each virus at least in triplicate. n indicates numbers of chickens used for infection by individual variants of pseudotyped viruses. Figure was created with Biorender.com.

All W38$^{-/-}$ chickens that were inoculated with pseudotypes with substitutions of arginine N311 developed tumors within 10 to 21 days. Of the six W38$^{-/-}$ chickens inoculated with the pseudotype carrying a S236L substitution, one developed a tumor 21 days after inoculation while the remaining five chickens remained healthy. Four W38$^{-/-}$ chickens were inoculated with the A432T pseudotype and all remained healthy. Out of the eight W38$^{-/-}$ chickens inoculated with a virus pseudotyped with the wt Env, one developed a tumor at day 35 (Fig 3A). The non-modified W38$^{+/+}$ chickens developed tumors after inoculation with all pseudotyped viruses (Fig 3B). Pseudotypes N311S, N311D as well as the wt induced tumor development within the first 10 days whereas pseudotypes S236L and A432T mutations induced smaller tumors within two to three weeks in the control W38$^{+/+}$ chicken. These results show the sensitivity of W38$^{-/-}$ chicken to two adapted virus variants with altered N311.

## NHE1 alleles with more extensive changes around W38 maintain the resistance to adapted Env variants

Having shown that ALV-J adapted to the single amino acid deletion, we were interested in whether the adapted virus variants gained the ability to infect the fibroblasts prepared from ALV-J-resistant species with more extensive changes in the critical region of NHE1. DF-1 cells with various sizes of deletions that were introduced by CRISPR/Cas9 and QT6 cells derived from Japanese quail were used for this experiment. All cells were infected with a virus carrying N311D and A432T substitutions. Selected variants represented viruses that differed in most of the tested parameters. GFP-positive cells were analyzed by flow cytometry 3 d.p.i. (Fig 4A). Fibroblasts prepared from chukar and satyr were sensitive to both tested adapted mutants, although significantly less sensitive than W38$^{-/-}$ CEF. The QT6 cells from Japanese quail and embryonic fibroblasts which were isolated from the gray partridge, guineafowl, and common pheasant were completely resistant to all variants.

DF-1 cells that have four amino acids deleted immediately before the preserved W38 (DF-1 Δ4) are resistant to wt RCASBP(J)GFP virus but highly sensitive to all adapted variants. In contrast, DF-1 cells with a 22 amino acid deletion that includes W38 (DF-1 Δ22) and DF-1 that possesses a frameshift mutation that leads to premature termination (DF-1 ΔNHE1) were resistant to all variants. This resistance again confirms that the adapted virus variants have not acquired specificity for a receptor molecule other than NHE1.

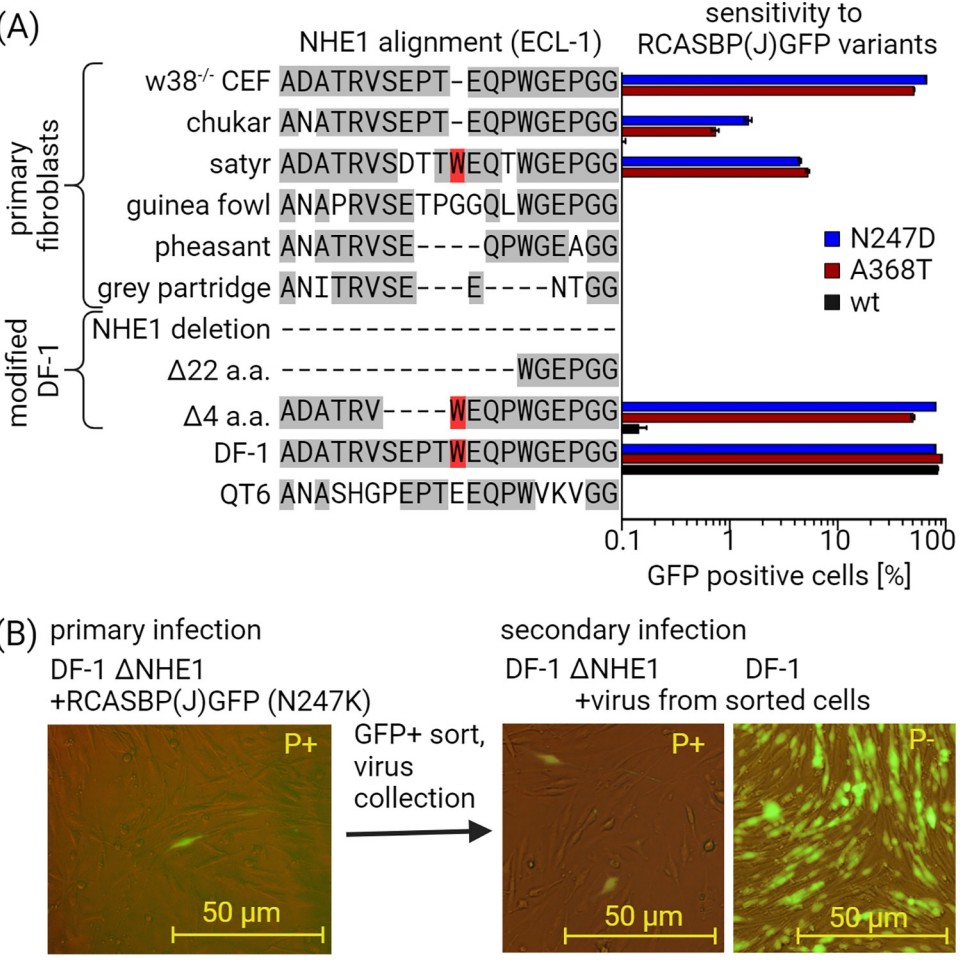

**Fig 4. Sensitivity of modified DF-1 cell lines and primary fibroblasts to RCASBP(J)GFP variants.** (A) Fibroblasts prepared from selected galliform species and DF-1 cell lines with partial or complete deletion of ALV-J receptor NHE1 were infected in triplicates by wt RCASBP(J)GFP and variants with mutations N311D and A432T. In the left column, the source of primary fibroblasts or NHE1 modification in DF-1 cells is described. The middle column shows alignment of the crucial part of the first extracellular loop of NHE1 molecule involved in ALV-J infection. Amino acids matching the sequence of the original DF-1 cell line are on a gray background. The W38 tryptophan is highlighted in red. Graph on the right side shows the percentage of GFP-positive cells analyzed by FACS 3 d.p.i. with the respective virus (mean ± SD). (B) Individual GFP-positive DF-1 ΔNHE1 cells infected with adapted RCASBP(J)GFP variants were sorted. Collected virus was used for secondary infection of DF-1 ΔNHE1 and DF-1. Representative photos of infection with a RCASBP(J)GFP variant carrying N311K substitution are shown. Other examples are shown in S2 Fig. Figure was created with Biorender.com.

In conclusion, these experiments indicate the relevance of the adapted viral variants in other cells that possess small changes in the critical region of NHE1. However, these adapted variants do not overcome the resistance of cells with extensive receptor modifications.

## Inefficient selection of Env variants adapted to NHE1 alleles with more extensive changes around W38

In order to assess the adaptability of Env-J to extended changes in the critical receptor region, we repeated the selection of virus variants that had already been adapted to W38[-/-] CEF using the DF-1 Δ22, DF-1 ΔNHE1 and QT6 cell lines (Figs 4B and S2). We performed the poly-brene-supported primary infection with all ΔW38-adapted variants and the wt RCASBP(J)

GFP. We observed individual GFP-positive cells in the infected cell cultures derived from the DF1 cell line. All infected QT6 cells were completely GFP-negative without any chance to sort infected cells and select adapted virus variants. It is of note that GFP-positive cells were not detected among those cells that were infected by wt virus and DF-1 ΔNHE1 infected by virus carrying the N311D substitution. As in the initial experiment, the GFP-positive cells were sorted, expanded, and cultured for virus collection and chromosomal DNA isolation (Fig 1A). The supernatant from the culture media was used for the infection of DF-1, DF-1 Δ4, DF-1 Δ22, and DF-1 ΔNHE1 cells. No increase of infection efficacy in DF-1 Δ22 and DF-1 ΔNHE1 cells occurred after secondary infection (Figs 4B and S2). As expected, the non-modified DF-1 and DF-1 Δ4 cells were efficiently infected after secondary infection. No new prevalent mutations were detected in the analyzed *env*-J sequences which have been illustrated by images of DF-1 and DF-1 ΔNHE1 infected with variants N311K, T313A, T313I, and T313N (Figs 4B and S2). Therefore, we conclude that the ΔW38-adapted virus variants are not able to adapt to the receptor with more extensive changes or switch to another receptor molecule.

## Discussion

In an experimental setting, genome editing using CRISPR/Cas9 was successfully used to induce resistance to infectious diseases in poultry. Minor mutations within the receptor or cofactor were shown to sufficiently induce chicken resistance to ALV-J and was recently also demonstrated with the avian influenza A virus [3,6]. In both cases, the experimental strategy was based on manipulating as little as possible the structure of proteins that exhibit important physiological functions. However, utilizing a strategy of introducing minor structural changes, may bring about a risk of the virus adaptation that would lead to the ability to bypass resistance. An example is highly pathogenic avian influenza A, which has recently been shown to overcome the resistance of genetically edited birds by adaptation of the viral polymerase to other cofactors from the ANP32 family [5,6].

In our study, we have performed a rapid adaptation of ALV-J to the chicken NHE1 receptor lacking W38. Eight different adapted virus variants with single amino acid substitutions replicating in CEF W38$^{-/-}$ were isolated. Six of these substitutions lead to the disruption of one N-glycosylation site (Fig 2A). Two variants, both with altered asparagine N311, were shown to induce tumors in an *in vivo* assay. Selected variants were also shown to overcome the natural resistance of some but not all galliform species *in vitro*.

Adaptation of ALV to new hosts has been the focus of many studies. In regards to this adaptation, it is generally accepted that Env is responsible for cell entry, therefore these types of studies have focused on the Env sequence. Modifications have been most frequently reported in the hr and vr regions of SU, which has been shown to be critical for receptor recognition [7,9,10,38–40] or at the C-terminal region of the SU glycoprotein and/or extracellular region of the TM glycoprotein, regions crucial for the viral membrane fusion process [41–43]. Selection of new ALV variants was supported by co-cultivation with resistant cells [9,10,40] either by passage through a resistant animal [41] or by addition of soluble Env [39,44], a soluble receptor [38,39,45] or inhibitor [42]. It is of note that the mechanism of host range extension of the J subgroup has not yet been studied. In contrast to the studies mentioned above, we obtained the adapted ALV-J variants in a relatively short time. We assumed that the adapted virus variants were already in the virus stock collected from the DF-1 cells. The mutant variants could be present at a sufficient titer in the DF-1 supernatant due to the high probability of a single amino acid substitution and equally high replication efficiency of all variants in the DF-1 cells (S2 Fig). Also, in contrast to most of the previously mentioned studies, we did not observe any cytotoxicity of the adapted virus variants. The high adaptation rate corresponds to

the minimum ΔW38 receptor modification. Polybrene was used to increase the infection efficiency by enhancing the virus adsorption on target cell membranes and facilitates both receptor-dependent and receptor-independent entry [46]. In our hands, polybrene increased the number of infected cells but did not change the spectrum of adapted virus variants (S1 Table).

Interestingly, although all obtained substitutions appeared rapidly in the cell culture, all sites altered in our adapted variants are invariable in all known strains of ALV-J [31,37]. In the case of the rare thermosensitive variant with an A432T substitution in HR1, this discrepancy can be explained by the fact that mutations in the HR1 domain often facilitate the fusion of the virus with the host cell but at the same time reduce the stability of the virus [41,42]. The other two loci could be important for protecting the virus against the host immune system [47,48]. In our experiments, only very young chickens (up to one month of age) were infected. This is key since it has been shown that the congenic chicken line, CB, which served as the source for establishing the W38[-/-] chickens, develops an effective adaptive immune response to ALV antigens only after 10 to 12 weeks of age [49]. Therefore, the immune system did not play a significant role in our experimental setup. Moreover, the disrupted N-glycosylation site was shown to be glycosylated only by complex forms of glycan chains [37], which are not recognized by the innate immune system [50]. In particular, according to the AlphaFold prediction, the N-glycosylation site N311 is on the very top of SU (S1 Fig) and its modification could influence both receptor binding as well as accessibility to neutralization antibodies and other components of adaptive immunity.

In contrast, the other two mutated residues are not protruding and probably are not involved directly in the receptor binding. Both S236L and A432T represent substitutions that change the hydrophobicity of the side chain, which could impact the protein folding and influence other steps of virus entry into the cell, such as the membrane fusion process. Mutations can also affect virus stability or protein export efficiency, but given the same viral replication efficiency in the DF-1 cell line, the mutations do not appear to have a significant effect on these properties *in vitro*.

All adapted variants of the virus tested *in vivo* induce tumors in wt W38[+/+] birds although more slowly than the wt virus. In W38[-/-] chickens, however, variants S236L and A432T are inefficient although they are well adapted *in vitro*. We can assume that virus-receptor interactions and subsequent fusion can be affected by a combination of factors, such as sensitivity to higher temperature *in vivo*, decreased virus stability, or sensitivity to lower pH, which may manifest in the acidic tumor environment. According to AlphaFold2 prediction, the substitutions S236L and A432T are not on the top of Env and not exposed for receptor binding. We therefore cannot neglect the possibility that substitutions S236L and A432T destabilize the Env structure and facilitate the release of fusion peptides, so that virus-cell fusion occurs even after suboptimal interaction with the receptor. Such destabilization and partial priming correlate with low thermostability and sensitivity to low pH [41], which may be a greater barrier in vivo than in vitro. Finally, the Env glycoproteins of adapted variants may present different antigenic epitopes causing slight variance in the immune response [49,51]. This can be sufficient to prevent tumor growth in W38[-/-] chicken compared with the situation in wt chickens.

Surprisingly, in one instance, a tumor was induced in a W38[-/-] chicken by the wt variant of the pseudotyped virus. This occurrence was an exception that did not appear in the previous study nor recurred in subsequent experiments. Unfortunately, due to the use of a pseudotyped virus, the explanation of the tumor induction is only hypothetical. We assume receptorless cell entry of the transforming virus and/or rescue of v-*src* transducing virus by an endogenous retrovirus of different subgroup specificity. Receptorless entry is also indicated by our *in vitro* experiments. (Figs 4B and S2). Using high-multiplicity infection, the receptorless virus entry was sporadically observed in vivo as well [41,52].

The adapted virus variants are also relevant in other naturally resistant species with minor receptor modifications. Here, we show that fibroblasts resistant to wt ALV-J prepared from chukar (W38 deletion) and satyr (3 amino acids substitutions, W38 preserved) can be infected with both tested adapted variants, while the other tested species with larger mutations remain resistant. Interestingly, although chukar has the same deletion as CEF W38$^{-/-}$ in a crucial part of the receptor, the sensitivity is almost two orders of magnitude lower (Fig 4A). Minor mutations in other parts of the receptor or some restriction factors present in chukar fibroblasts may play a role. The inability of the virus to adapt to more extensive modifications of the receptor was confirmed on the modified DF-1 cells. After long-term cultivation and selection of all adapted virus variants and the wt virus in resistant DF-1 Δ22 and DF-1Δ NHE1, we did not obtain a single adapted virus. All observed GFP-positive cells were apparently the result of the non-specific receptor independent virus entrance facilitated by polybrene (Fig 4B).

Our finding that larger receptor modifications can ensure quite stable resistance is consistent with a recent study of biotechnologically induced resistance to avian influenza in chickens [6]. Both studies clearly demonstrated the need for complex gene editing in an attempt to create genetic resistance to viral diseases and the importance of performing a thorough analysis to assess the potential impact of genome-edited livestock on virus evolution. Based on this study, we propose to create chickens resistant to ALV-J by editing the NHE1 receptor sequence based on other related resistant galliforms with more extended receptor modifications. This could ensure both fitness preservation and stable resistance. Proper *in vitro* testing of resistance stability should precede the genetic modification of the chickens *in vivo*. This approach aims to minimize the risk of the emergence of adapted mutants.

## Materials and methods

### Ethics statement

We conducted all experiments and procedures in accordance with the Czech legislation for animal handling and welfare. The Animal Commodities Department of the Ministry of Agriculture of the Czech Republic approved all animal experiments described in this study (approval no. 31255/2019-MZE-18134).

### Experimental animals and breeding

ΔW38 chicken line was prepared as described previously [3] and has since been maintained at the Institute of Molecular Genetics, Czech Academy of Sciences, Prague, Czech Republic in heterozygous W38$^{+/-}$ state. W38$^{-/-}$ and W38$^{+/+}$ embryos or chickens used for experimental infections were obtained from crossing heterozygous parents. Standard housing conditions (16 h light/8 h dark cycle and food/water provided ad libitum) were applied. Eggs were incubated in a forced air incubator (BIOS MIDI) under standard hatching conditions.

### Embryonic fibroblasts and cell line cultivation

Chicken W38$^{-/-}$, chicken W38$^{+/+}$, chukar partridge (*Alectoris chukar*), satyr tragopan (*Tragopan satyra*), Guineafowl (*Numida meleagris*), common pheasant (*Phasianus colchicus*) and gray partridge (*Perdix perdix*) embryonic fibroblasts were prepared from 10-day-old embryos as described previously [53]. All embryonic fibroblasts, chicken permanent cell line DF-1, and Japanese quail QT6 cells were grown in a mixture of two parts of DMEM and one part of F-12 medium supplemented with 5% calf serum, 5% fetal calf serum, 1% chicken serum, and penicillin/streptomycin (100 μg/mL each) in a 5% $CO_2$ atmosphere at 37˚C.

### Selection and sequencing of adapted virus variants

The RCASPB(J)GFP virus [24,54] was used for infection with MOI 5.0 with or without polybrene at a concentration of 8 µg/mL of media. W38$^{-/-}$ CEF, DF-1 Δ4, DF-1 Δ22, and DF-1 ΔNHE1 were used for selection. Clones DF-1 Δ4, DF-1 Δ22, and DF-1 ΔNHE1 were denoted as 4–22, 4–9 and 4–1 in the previous study [23]. Virus infection and spread were observed as an increasing proportion of GFP-positive cells using fluorescence microscopes (Olympus IX5 and Leica DMi8) and quantified by flow cytometry (BD LSRII). After 1–2 weeks of cultivation, GFP-positive cells were sorted by flow cytometry. The pool of GFP-positive cells was expanded, medium with selected virus was harvested, cleared of debris by syringe filters with 0.45 µm membrane (Corning), and stored as aliquots at −80˚C. The harvested virus was used for infection of cells with MOI 5.0 with or without polybrene at a concentration of 8 µg/mL of media. Genomic DNA was isolated by phenol–chloroform extraction. The RCASBP(J)GFP *env* gene was amplified using the forward primer Pol_F (5′-GGGGAGTGGGAAAAAGGATGGAAC-3′) and reverse primer GFP_R (5′- GGCCA CAGTGGTCTAGAATCGATG-3′). The following PCR conditions were used: 98˚C for 30 s, 35 cycles of 98˚C for 10 s, 66˚C for 30 s, and 72˚C for 2 min, and terminal extension at 72˚C for 5 min with Phusion polymerase (NEB). The resulting PCR product of 2409 bp in length was sequenced from both sides with the primers used in the PCR amplification and with one additional inside primer Env_F (5'- CCTTTGGTTCGGTGTGCTA-3′).

### Virus propagation

The stocks of RCASBP(J)GFP virus and its *env* variants with substitutions S236L, N311S, N311D, N311K, T313A, T313I, T313N, and A307T used in this study were propagated by transfection of the vector plasmid DNA into DF-1 cells using the Lipofectamine 3000 Transfection Reagent (Invitrogen), harvested on day 7 post transfection, cleared of debris by syringe filters with 0.45 mm membrane (Corning) and stored as aliquots at −80˚C. The virus titer was determined by terminal dilution and subsequent infection of DF-1 cells as $10^6$ IU/mL. The transforming virus of subgroup J for *in vivo* sarcoma induction was produced by rescuing replication-defective BH-RSV present in the 16Q cell line [55]. 16Q cells were transfected with plasmid RCASBP(J)GFP or its variants carrying substitutions S236L, N311S, N311D, or A307T using Lipofectamine 3000 Transfection Reagent. The virus stock containing both GFP-reporter viruses and transforming viruses pseudotyped with respective variants of Env-J was harvested on day 3 post transfection, cleared of debris by syringe filters with 0.45 µm membrane and stored as aliquots at −80˚C. The titers of the transforming virus RCASBP(J)GFP and its variants with substitutions S236L, N311S, N311D, or A307T were determined by focus assay in Brown Leghorn CEF as $9x10^4$, $5x10^4$, $8x10^4$, $5x10^4$, and $2x10^4$ FFU/mL, respectively.

### Cloning of virus variants

The detected substitutions were introduced into the RCASBP(J)GFP plasmid using the In-Fusion Cloning Kit (TaKaRa). Fragments for cloning were prepared with Phusion polymerase (NEB) in the following conditions: 98˚C for 30 s, 30 cycles of 98˚C for 10 s, 65˚C for 30 s, and 72˚C for 40 s and terminal extension at 72˚C for 5 min. The resulting fragments were mixed according to the protocol. All primers used for in-fusion cloning of new virus variants are described in supplementary information (S4 Table).

### Cloning of RCASBP(J)dsRED

The RCASBP(J)dsRed vector was prepared by replacing the *gfp* gene of RCASBP(J)GFP with *dsRed-Monomer*. First, the *Cla*I site within *env*-J and the *gfp* gene was removed from the RCAS

(J)GFP vector. Forward 5'-GCAGAATCGATCAGCCATTGATTTCTTACTC-3' and reverse 5'-TGGCCCGTACATCGCATCGATGCCACAGTGGTACGC-3' primers were used for amplification from RCASBP(J)GFP. The resulting PCR product was then cloned into the 11 kb *Cla*I fragment of RCASBP(J)GFP using the In-Fusion Cloning Kit. The substituted base in the primer is capitalized; the site of deletion in the primer is marked with a dash. Next, dsRed-Monomer was cloned into the *Cla*I-linearized RCASBP(J) vector as previously described [13].

### *In vivo* tumor induction in chickens

W38[+/+] and W38[-/-] chickens [3] at the age of 19 to 34 days were inoculated with 250 or 500 FFU of transforming virus rescued from 16Q cells in 0.1 mL of PBS into the left and right breast muscle, respectively. The growth of sarcomas at the site of inoculation was monitored (S3 Fig). When the tumor was detected, the bird was euthanized. Tumors were fixed in 4% buffered paraformaldehyde, embedded in paraffin blocks, and processed by routine histological procedures with haematoxylin and eosin staining.

### Structural analyses

The trimer of Env-J protein was modeled using AlphaFold [56] available on Google Colaboratory server [57]. Molecular graphics and analyses were performed with UCSF ChimeraX software, version 1.7.dev202310120058 [58].

### DNA sequence selection analysis

One hundred ninety-seven publicly available DNA sequences of *env*-J (S1 Dataset) were aligned as translated amino acids using the Muscle algorithm. Final alignment was manually inspected and edited. To test the presence of natural selection in the *env*, we used the FEL (Fixed Effects Likelihood) method [59] implemented in the HyPhy v.2.5.33 package [60] with default parameters. As a result, estimates of synonymous (dS) and non-synonymous (dN) substitution rates for each site were obtained. The presence of natural selection pressure was evaluated using the Likelihood Ratio Test (LRT) testing the hypothesis that dS is not equal to dN.

### Supporting information

**S1 Table. List of experimental infections of resistant cells and mutations found in env glycoprotein of selected virus stocks.** Independent DF-1 transfections are indicated by letters A to I, individual virus stocks are indicated by numbers. Presence (+) or absence (-) of polybrene is indicated in a column marked with letter P.
(XLSX)

**S2 Table. Significance of differences in replication efficiency of ALV-J variants in W38[-/-] CEF (Fig 2B).** Significance is statistically analyzed by t-test. Significant differences are marked by asterisks (*P<0.05, **P<0.01, ***P < 0.001).
(XLSX)

**S3 Table. Significance of differences in virus spread of ALV-J variants in DF-1 at 42˚C at the day 4 of co-cultivation (Fig 2E).** Significance is statistically analyzed by t-test. Significant differences are marked by asterisks (*P<0.05, **P<0.01, ***P < 0.001).
(XLSX)

**S4 Table. List of oligonucleotide primers used for introduction of substitutions into the RCASBP(J)GFP vector.** The substituted base is capitalized.
(XLSX)

**S1 Fig. Structure of ALV-J envelope trimers predicted by AlphaFold with marked sites of detected adaptive substitutions and disrupted glycosylation site.** Each monomer is marked with a different color. Envelope is displayed from the side view as (A) a cartoon with cylinders and stubs, (B) with apparent molecular surface, and (C) from the top view with apparent molecular surface. Figure was created with Biorender.com.
(TIF)

**S2 Fig. Selection of adapted viruses on DF-1 ΔNHE1 cells.** Individual GFP-positive DF-1 ΔNHE1 cells infected with adapted RCASBP(J)GFP variants were sorted. Collected virus stocks were used for secondary infection of DF-1 ΔNHE1 and DF-1. Representative photos of infection with RCASBP(J)GFP variants carrying the T313A, T313I, or T313N substitution are shown. Infection with other adapted virus variants carrying substitutions S236L, N311S, N311D, and N313T and with other cells (DF-1 Δ22 and DF-1 Δ4) were performed as well. Figure was created with Biorender.com.
(TIF)

**S3 Fig. Representative sarcoma induced at the site of v-*src* transducing virus inoculation.** (A) Sarcoma *in situ* in the left pectoral muscle and (B) haematoxylin and eosin staining of a tumor section. Figure was created with Biorender.com.
(TIF)

**S1 Dataset.** **(Sheet A)** NCBI accession numbers. **(Sheet B)** Amino acid variability of Env-J sequences obtained from databases and type of selection pressure applied to individual amino acids calculated based on the ratio of non-synonymous to synonymous substitutions (dN/dS). The positions of amino acids substituted in our experiments are highlighted in gray, the start of the TM is indicated by a double line.
(XLSX)

## Acknowledgments

We thank Zdeněk Cimburek, Filip Šenigl, and Eliška Gáliková (all from Institute of Molecular Genetics, Prague) for their kind help with FACS, cell sorting, and fluorescence microscopy experiments. Pavel Trefil and Jitka Mucksová (BIOPHARM, Research Institute of Biopharmacy and Veterinary Drugs, Jílové, Czech Republic) helped us with breeding the ΔW38 chicken line. We also thank Katerina Trejbalová and Daniel Elleder (Institute of Molecular Genetics, Prague) for critical reading and comments on the manuscript, and Šárka Takáčová and Jasper Manning (both from Institute of Molecular Genetics, Prague) for native speaker proofreading the manuscript.

## Author Contributions

**Conceptualization:** Magda Matoušková, Jiří Hejnar.

**Data curation:** Magda Matoušková.

**Formal analysis:** Magda Matoušková, Jiří Hejnar.

**Funding acquisition:** Jiří Hejnar.

**Investigation:** Magda Matoušková, Jiří Plachý, Dana Kučerová, Ľubomíra Pecnová, Markéta Reinišová, Josef Geryk, Tomáš Hron, Jiří Hejnar.

**Methodology:** Jiří Plachý, Dana Kučerová, Ľubomíra Pecnová, Markéta Reinišová, Josef Geryk, Vít Karafiát, Tomáš Hron.

**Resources:** Jiří Hejnar.

**Supervision:** Jiří Hejnar.

**Visualization:** Magda Matoušková.

**Writing – original draft:** Magda Matoušková.

**Writing – review & editing:** Jiří Plachý, Jiří Hejnar.

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
