## [Decision Letter · Decision Letter 0]

27 May 2024

Dear Hejnar,

Thank you very much for submitting your manuscript "Rapid adaptive evolution of avian leukosis virus in response to biotechnologically induced host resistance" for consideration at PLOS Pathogens. As with all papers reviewed by the journal, your manuscript was reviewed by members of the editorial board and by several independent reviewers. The reviewers appreciated the attention to an important topic. Based on the reviews, we are likely to accept this manuscript for publication, providing that you modify the manuscript according to the review recommendations.

All reviewers found your results of interest and importance. Reviewer 1 has suggested a number of additional experiments: if you consider that these are not essential for the purposes of the present paper, then please provide a full justification.

In addition, the reviewers - particularly reviewer 3 - make a number of suggestions for the phrasing of the paper and the interpretation of your data. Please consider these carefully and indicate your response to each point.

Sincerely,

Charles R M Bangham, ScD FRS

Academic Editor

PLOS Pathogens

Richard Koup

Section Editor

PLOS Pathogens

Michael Malim

Editor-in-Chief

PLOS Pathogens

orcid.org/0000-0002-7699-2064

All reviewers found your results of interest and importance. Reviewer 1 has suggested a number of additional experiments: if you consider that these are not essential for the purposes of the present paper, then please provide a full justification.

In addition, the reviewers - particularly reviewer 3 - make a number of suggestions for the phrasing of the paper and the interpretation of your data. Please consider these carefully and indicate your response to each point.

Reviewer Comments (if any, and for reference):

Reviewer's Responses to Questions

**Part I - Summary**

Reviewer #1: This study used cell lines lacking W38 NHE1 to identify adaptive mutations within the viral envelope protein of ALV-J, demonstrating that changes in amino acids in the envelope protein can lead to replicate effectively in W38-/- chicken embryonic fibroblasts in vitro while in vivo. Consequently, it underscores the necessity for more intricate genomic editing to establish enduring antiviral defenses in poultry. The study contains interesting information for biotechnologically induced host resistance but in my opinion has important issues that have not been adequately addressed in the manuscript. The current results do not fully support the conclusion.

Reviewer #2: This ms. represents a logical extension of previous work in which this laboratory produced genetically modified chickens that have altered receptors that cause resistance to the pathogenic subgroup J ALVs. Birds were produced with different degrees of alteration in the subgroup J receptor (the NHE1 gene). The results are clear and for the most part what could have been predicted based on what is known about retroviruses in general and ALVs in particular. In tissue culture experiments done in cells from birds in which the modification is minimal (a single amino acid deleted) the virus readily adapted through mutations in the env gene. Most of the adaptive changes occurred near, but not in, the canonical variable regions that have been associated with host range/receptor choice in the ALV env gene. There are some interesting complexities when the in vitro data are compared with data from a limited set of experiments done in birds that the authors should explore more fully in the ms.

Reviewer #3: Matouskova et al present a very nicely written and well executed manuscript about the selection of mutants of ALV-J that emerge in gene edited chicken cells W38-/- that express a mutated viral receptor protein lacking a key amino acid (W38) that the ALV-J env interacts with for entry.

The selection has been performed in vitro using CRISPR gene edited cells that harbour the NHE-1 gene with a single amino acid W38 deleted. The virus populations that evolve in these cells contain a variety of different mutations , eight common ones in replicate evolution experiments suggesting common pathways to resistance.

The mutated envs allow entry into cells from other species that have natural variation at this position and also enable infection of chickens and induction of tumours.

The final conclusion is that the single amino acid deletion is not enough to prevent emergence of virus resistance and more extensive edits are required, bearing in mind always though that there is balance between disrupting host gene function and preventing virus replication.

The data are all presented in appropriate ways using good number of repeats and statistics. Virus growth in wild type or -/- cells is recorded largely by measuring GFP positive cell numbers using a reporter virus that expresses GFP. Results are always expressed as percentages either of total cells numbers or ratios compared to wild type.

The only request I have to improve the readability of this manuscript is that the basic systems should be better explained up front. It is quite hard work to understand exactly what the viruses/ reporter systems used here are and what the assays are too. Figure 1 is a welcome schematic, but what IS the RCASBGFP vector? To readers who are not familiar with these surrogate systems this needs explaining in the text at the start of the result section. Similarly the complementation assay presented in figure 2F is very nice evidence that the mutated envs do not use alterative receptor but still rely on NHE-1. This assay is very important because the ALV family switch receptors during diversifying evolution. However the text describing this assay is minimal and the reader has to work hard to understand that a wild type reporter expressing ds red has been generated and used in coinfection studies, please embellish the text within the results section to explain so the reader does not have to interrupt the flow but cross referencing back and forth into methods section.

Alphafold is used to show the position of the mutations in emergent viruses supp figure S1. Residues 311 and 313 are clearly poised to interact directly with the NHE-1 receptor. However, some other mutations are not- this suggests that increased protein expression transport or stability might also confer some way to infect cells with the suboptimal receptor- did the authors check for protein expression of the mutated envs?

In figure 3 chickens are infected with the virus and tumours are induced even by wild type vbirus in W38-/- chickens Did the authors sequence virus in these tumours after day 30? ASdditioally did the authors sequence virus in chickens infected with other mutants in which high numbers of tumours were present? Perhaps further mutations have appeared?

Minor points:

1. In abstract remove Only from tumor induction by only two of the variants, since not all variants were tested.

2. Please insert a label for x axis (days?) in figure 3 A and B.

**Part II – Major Issues: Key Experiments Required for Acceptance**

Reviewer #1: 1. For the eight single nucleotide substitutions discovered, is it possible to conduct further bioinformatics analysis to predict the potential impact of these mutations on viral function and structure?

2. Have you considered the potential for these adaptive variants to spread under natural conditions and their impact on poultry populations? In other words, is it currently prevalent for these adaptive mutants within the chicken population?

3. The conclusion drawn from result 1 is inappropriate, as the data in this section only counted the number of GFP positive cells, which is insufficient to confirm the claim that a single amino acid mutation confers adaptation of the virus to host resistance. The expression level and viral titer of viral proteins (p27 or/and gp85) should also be tested.

4. Fig2. Replication efficiency of ALV-J variants should be test by other methods, such as viral protein (p27 or/and gp85) expression level of ALV-J, except for the GFP positive cell percentage.

5. In Figure 2, the replication efficiency of the ALV-J variant should be tested using other methods, such as the expression levels of ALV-J viral proteins (p27 or/and gp85), besides the percentage of GFP positive cells.

6. In Figure 3a, please provide additional details regarding this experimental design in the M&M section. What types of tumors have been induced? In which organs did tumors occur? Images of histopathological changes in the tumor tissue should be provided.

7. To assess the role of detected eight mutations (S236L, N311S, N311D, N311K, T313A, T313I, T313N, and A432T), each of them was only introduced into the RCASBP(J)GFP vector. These results do not fully support this conclusion of single amino acid substitutions within Env-J are sufficient enough to circumvent ΔW38-induced resistance. These eight mutations should be introduced into ALV-J, such as the prototype strain HPRS-103, to further validate their ability to infect W38-/- CEFs.

Reviewer #2: 1) Given that the obvious goal was to develop resistant chickens, and that the authors are clearly aware that the body temperature of the chickens is significantly higher than 37, which can affect the replication of the mutant viruses (as shown in figure 2), it is not obvious why the selection experiments were done at 37 and not at 42.

2) The selections and most of the analysis were done in cultured cells, which is both experimentally simpler and cheaper than in vivo experiments. As shown in figure 3, the in vivo infectivity data in the mutant birds are a better match for the ability of the mutant viruses to grow in tissue culture at 42 than at 37. What is more surprising is that the WT virus produced a tumor in what should have been a stringent in vivo infectivity assay. In that sense, based on an experiment with a small number of birds, the WT virus appeared to be more infectious for the mutant birds in vivo than were several of the mutants that were selected in tissue culture. In addition, all of the mutant viruses caused tumors in WT birds (although more slowly than WT), which means that the higher temperature in vivo cannot wholly explain the results presented in figure 3. The authors don’t comment on either of these unexpected results; they need to discuss the unexpected nature of the data shown in both panels of figure 3.

3) The naïve reader may not understand the basis of the in vivo tumorigenesis experiments based on how these experiments are described in the text. The env gene is deleted in the BH-RSV strain. In the in vivo experiments, the RCASBP(J) viruses (WT and mutant) act as a helper virus and supply the env that makes the BH-RSV infectious. Although the resulting BH-RSV is a one round virus, the presence of the RCASBP(J) in the stock means that a cell in a bird that is infected with BH-RSV could be reinfected with the helper virus and, if that happens, BH-RSV can be mobilized in the bird. That could help to explain the difference in the time it takes to see a visible tumor in the experiments shown in figure 3B. Because the helper virus can replicate, there is also the possibility that mutant/revertant viruses could have arisen in some of the in vivo experiments. That possibility suggests that different results might have been obtained if the birds were inoculated either with RCASBP(J) variants that carry src (or some other marker), which would have made it a simple matter to look for env mutants/revertants. Alternatively, if BH-RSV stock had been prepared by transfection with an env expressing plasmid and not a replication competent helper, the BH-RSV virus should not have been able to replicate, or revert/mutate. Whatever the explanation for the data in figure 3B, the nature of the BH-RSV experiments, and the potential implications of the ability of the helper viruses not only to provide an infectious env, but also to replicate, should be discussed.

4) As illustrated by comments 2 and 3, the Discussion does not come to grips with some of the complexities presented by the data in the ms. In fact, much of what is in the current version of the Discussion is either already presented in, or should be presented in, the Introduction. The Discussion should focus on what the new data in the ms. show. One possibility would be to create a combined Results and Discussion, so that the average reader, who is unlikely to bring much knowledge of the complexities of ALV biology to the ms., will be able to properly appreciate why they experiments were done the way they were, and what the data mean.

Reviewer #3: No more experiments required. add data on virus sequence in chickens if possible

**Part III – Minor Issues: Editorial and Data Presentation Modifications**

Reviewer #1: 1. The scope of the article title is too broad, and only rapid adaptive evolution of ALV-J was detected in the article, without detecting other subgroups, so avian leukosis virus should not be used.

2. Please provide additional details regarding the experimental design in the M&M section, particularly the process of screening and identifying adapted viral variants. For example, why choose to extend the cultivation period by two weeks instead of longer? Will longer cultivation periods produce more adaptive virus mutants?

Reviewer #2: none

Reviewer #3: Minor points:

1. Please give more detail in text about the reporter virus system used here and the complementation assay

2. In abstract remove Only from tumor induction by only two of the variants, since not all variants were tested.

3. Please insert a label for x axis (days?) in figure 3 A and B.

PLOS authors have the option to publish the peer review history of their article (what does this mean?). If published, this will include your full peer review and any attached files.

Reviewer #1: No

Reviewer #2: No

Reviewer #3: No

Figure Files:

Data Requirements:

Reproducibility:

References:

---

## [Editor Report · Decision Letter 1]

29 Jul 2024

Dear Prof. Hejnar,

We are pleased to inform you that your manuscript 'Rapid adaptive evolution of avian leukosis virus subgroup J in response to biotechnologically induced host resistance' has been provisionally accepted for publication in PLOS Pathogens.

Best regards,

Charles R M Bangham, ScD FRS

Academic Editor

PLOS Pathogens

Richard Koup

Section Editor

PLOS Pathogens

Michael Malim

Editor-in-Chief

PLOS Pathogens

orcid.org/0000-0002-7699-2064
---

## [Editor Report · Acceptance letter]

11 Aug 2024

Dear Prof. Hejnar,

We are delighted to inform you that your manuscript, "Rapid adaptive evolution of avian leukosis virus subgroup J in response to biotechnologically induced host resistance," has been formally accepted for publication in PLOS Pathogens.

Best regards,

Michael Malim

Editor-in-Chief

PLOS Pathogens

orcid.org/0000-0002-7699-2064